# Improving Encoder-Decoder CNN for Inverse Problems

## Abstract

Encoder-decoder convolutional neural networks (CNN) have been extensively used for various inverse problems. However, their prediction error for unseen test data is difficult to estimate a priori, since the neural networks are trained using only selected data and their architectures are largely considered blackboxes. This poses a fundamental challenge in improving the performance of neural networks. Recently, it was shown that Stein's unbiased risk estimator (SURE) can be used as an unbiased estimator of the prediction error for denoising problems. However, the computation of the divergence term in SURE is difficult to implement in a neural network framework, and the condition to avoid trivial identity mapping is not well defined. In this paper, inspired by the finding that an encoder-decoder CNN can be expressed as a piecewise linear representation, we provide a close form expression of the unbiased estimator for the prediction error. The close form representation leads to a novel bootstrap and aggregation scheme to prevent a neural network from converging to an identity mapping so that it can enhance the performance. Experimental results show that the proposed algorithm provides consistent improvement in various inverse problems.

## 1 Introduction

Suppose that the measurement $x$ is corrupted with additive noises:

$$x = \mu + w, \quad w \sim \mathcal{N}(0, \sigma^2 I), \tag{1}$$

where $\mu \in \mathbb{R}^n$ denotes the unknown mean vector, and $\sigma^2$ is the variance. Consider a deep neural network model $F(x) := F_\Theta(x)$ with the weight $\Theta$, which is trained with the data $x$ producing an estimate $\widehat{\mu} = F_\Theta(x)$. Then, our goal is to estimate the *prediction error*:

$$\text{Err}(\widehat{\mu}) = E_{x,x^*} \|x^* - \widehat{\mu}\|^2 = E_{x,x^*} \|x^* - F(x)\|^2, \tag{2}$$

which quantifies how well $\widehat{\mu}$ can predict a test data $x^*$, independently drawn from the same distribution of $x$ (Efron, 2004; Tibshirani & Rosset, 2018). The problem of estimating the prediction error is closely related to the generalizability of neural network (Anthony & Bartlett, 2009). Moreover, this problem is tightly linked to the classical approaches for model order selection in statistics literature (Stoica & Selen, 2004). One of the most investigated statistical theories to address this question is so-called covariance penalties approaches such as Mallow's Cp (Mallows, 1973), Akaike's information criterion (AIC) (Akaike, 1974), Stein's unbiased risk estimate (SURE) (Donoho & Johnstone, 1995), etc.

This paper is particularly interested in estimating the prediction error of encoder-decoder convolutional neural networks (E-D CNNs) such as U-Net (Ronneberger et al., 2015; Han & Ye, 2018; Ye & Sung, 2019). E-D CNNs have been extensively used for various inverse problems (Ye et al., 2018). Recent theoretical results showed that, thanks to the ReLU nonlinearities, the input space is partitioned into non-overlapping regions so that input images in each region share the same linear representation, but not across different partitions (Ye & Sung, 2019).

One of the most important contributions of this paper is that this property can be exploited to derive a simplified form of prediction error estimator that can be used for neural network training. In particular, we provide a close form expression for the divergence term in the SURE estimator, which

suggests that the divergence term can be neglected if a proper batch normalization is used. Furthermore, the simplified form of the SURE estimate for the prediction error leads to a bootstrap and aggregation scheme to prevent the CNN from converging to a trivial solution so that it can improve the neural network performance. The applicability of the new method is demonstrated using various inverse problems such as accelerated MRI, energy-dispersive X-ray spectroscopic imaging, deconvolution, and etc, which clearly show that our method can significantly improve the image quality compared to the existing approaches.

## 2   RELATED WORKS

### 2.1   STEIN RISK UNBIASED ESTIMATION FOR PREDICTION ERROR

The Stein Risk Unbiased Estimator (SURE) for the prediction error can be represented by (Donoho & Johnstone, 1995):

$$\widehat{\mathrm{SURE}}(\boldsymbol{x}) := \|\boldsymbol{x} - \boldsymbol{F}(\boldsymbol{x})\|^2 + 2\sigma^2 \mathrm{div}\{\boldsymbol{F}(\boldsymbol{x})\}, \tag{3}$$

where $\mathrm{div}(\cdot)$ denotes the divergence. Recently, the authors in (Soltanayev & Chun, 2018) employed the SURE for CNN-based image denoising without references by minimizing the following loss:

$$\ell_{SURE}(\boldsymbol{\Theta}) := \frac{1}{P} \sum_{i=1}^{P} \|\boldsymbol{x}^{(i)} - \boldsymbol{F}_{\boldsymbol{\Theta}}(\boldsymbol{x}^{(i)})\|^2 + 2\sigma^2 \mathrm{div}\{\boldsymbol{F}_{\boldsymbol{\Theta}}(\boldsymbol{x}^{(i)})\}. \tag{4}$$

where $\{\boldsymbol{x}^{(i)}\}_{i=1}^{P}$ denotes the training samples. Although the application of SURE for unsupervised denoising is an important advance in theory, there are several practical limitations. In particular, due to the difficulty of calculating the divergence term, the authors relied on MonteCarlo SURE (Ramani et al., 2008) which calculates the divergence term using MonteCarlo simulation. This introduces additional hyperparameters, on which the final results critically depend.

### 2.2   PIECEWISE LINEAR REPRESENTATION BY E-D CNN WITH RELUS

Recently, the authors in (Ye & Sung, 2019) showed that the output of a CNN composed of $\kappa$ layer of encoder and decoder without skipped connection can be represented by the following basis-like representation:

$$\boldsymbol{y} := \boldsymbol{F}(\boldsymbol{x}) = \sum_{i} \langle \boldsymbol{b}_i(\boldsymbol{x}), \boldsymbol{x} \rangle \tilde{\boldsymbol{b}}_i(\boldsymbol{x}) = \tilde{\boldsymbol{B}}(\boldsymbol{x}) \boldsymbol{B}(\boldsymbol{x})^\top \boldsymbol{x}, \tag{5}$$

where $\boldsymbol{b}_i(\boldsymbol{x})$ and $\tilde{\boldsymbol{b}}_i(\boldsymbol{x})$ denote the $i$-th column of the following frame basis and its dual:

$$\begin{aligned} \boldsymbol{B}(\boldsymbol{x}) &= \boldsymbol{E}^1 \boldsymbol{\Sigma}^1(\boldsymbol{x}) \boldsymbol{E}^2 \cdots \boldsymbol{\Sigma}^{\kappa-1}(\boldsymbol{x}) \boldsymbol{E}^\kappa, &(6) \\ \tilde{\boldsymbol{B}}(\boldsymbol{x}) &= \boldsymbol{D}^1 \tilde{\boldsymbol{\Sigma}}^1(\boldsymbol{x}) \boldsymbol{D}^2 \cdots \tilde{\boldsymbol{\Sigma}}^{\kappa-1}(\boldsymbol{x}) \boldsymbol{D}^\kappa, &(7) \end{aligned}$$

where $\boldsymbol{\Sigma}^l(\boldsymbol{x})$ and $\tilde{\boldsymbol{\Sigma}}^l(\boldsymbol{x})$ denote the diagonal matrix with 0 and 1 values that are determined by the ReLU output in the previous convolution steps. $\boldsymbol{E}^l$ and $\boldsymbol{D}^l$ refer to the encoder and decoder matrices, respectively, which are determined by pooling (resp. unpooling) operation and convolution filters (Ye & Sung, 2019). For the case of encoder-decoder CNN with the skipped connection, similar basis expression can be obtained Ye & Sung (2019).

Note that the expression (5) is *nonlinear* due to the dependency on the input signal $\boldsymbol{x}$. Moreover, as the ReLU nonlinearity is applied after the convolution operation, the on-and-off activation pattern of each ReLU determines a binary partition of the feature space across the hyperplane that is determined by the convolution. Accordingly, one of the most important observation in Ye & Sung (2019) is that the input space $\mathcal{X}$ is partitioned into multiple non-overlapping regions so that input images for each region share the same linear representation, but overall representation is still non-linear. This implies that two different input images may be automatically switched to two distinct linear representations that are different from each other.

## 3 MAIN CONTRIBUTION

In (3) and (4), the calculation of the divergence term is not trivial for general neural networks. This is why the authors in (Soltanayev & Chun, 2018) employed the MonteCarlo SURE. Thanks to the basis-like expression in (5), here we show that there exists a simple explicit form of the divergence term for the case of E-D CNNs. Then, the batch normalization is shown to make the divergence term trivial.

### 3.1 DIVERGENCE SIMPLIFICATION IN SURE

In Proposition 6 of (Ye & Sung, 2019), the authors show that $\partial \boldsymbol{F}(\boldsymbol{x})/\partial \boldsymbol{x} = \tilde{\boldsymbol{B}}(\boldsymbol{x})\boldsymbol{B}(\boldsymbol{x})^\top$ for the case of E-D CNN with ReLUs. Accordingly, we have

$$
\begin{aligned}
\text{div}\{\boldsymbol{F}(\boldsymbol{x})\} = \text{Tr}\left(\frac{\partial \boldsymbol{F}(\boldsymbol{x})}{\partial \boldsymbol{x}}\right) &= \text{Tr}\left(\tilde{\boldsymbol{B}}(\boldsymbol{x})\boldsymbol{B}(\boldsymbol{x})^\top\right) = \text{Tr}\left(\boldsymbol{B}(\boldsymbol{x})^\top\tilde{\boldsymbol{B}}(\boldsymbol{x})\right) \\
&= \sum_i \langle \boldsymbol{b}_i(\boldsymbol{x}), \tilde{\boldsymbol{b}}_i(\boldsymbol{x})\rangle,
\end{aligned} \tag{8}
$$

where $\text{Tr}(\boldsymbol{A})$ denotes the trace of $\boldsymbol{A}$.

This close form representation (8) leads to a further simplification of divergence term by exploiting the property of the batch normalization. Recall that batch normalization has been extensively used to make the training stable (Ioffe & Szegedy, 2015; Hoffer et al., 2018; Cho & Lee, 2017; Miyato et al., 2018; Ulyanov et al., 2016). It has been consistently shown that the batch normalization is closely related to the norm of the Jacobian matrix $\partial \boldsymbol{F}(\boldsymbol{x})/\partial \boldsymbol{x}$, which is equal to $\tilde{\boldsymbol{B}}(\boldsymbol{x})\boldsymbol{B}(\boldsymbol{x})^\top$. For example, in their original paper (Ioffe & Szegedy, 2015), the authors conjectured that "Batch Normalization may lead the layer Jacobians to have singular values close to 1, which is known to be beneficial for training". By extending the idea in (Ioffe & Szegedy, 2015) to multiple layers, the batch normalization can be understood as to make the covariance of the network output and input similar. For example, for the uncorrelated input with $\text{Cov}[\boldsymbol{x}^{(i)}] = \sigma^2 \boldsymbol{I}$, the batch normalization may result in $\text{Cov}[\boldsymbol{F}(\boldsymbol{x}^{(i)})] \simeq \sigma^2 \boldsymbol{I}$. Furthermore, for sufficiently smaller $\sigma$, we have

$$
\begin{aligned}
\sigma^2 \boldsymbol{I} \simeq \text{Cov}[\boldsymbol{F}(\boldsymbol{x}^{(i)})] &= \text{Cov}\left[\tilde{\boldsymbol{B}}(\boldsymbol{x}^{(i)})\boldsymbol{B}(\boldsymbol{x}^{(i)})^\top \boldsymbol{x}^{(i)}\boldsymbol{x}^{\top(i)}\boldsymbol{B}(\boldsymbol{x}^{(i)})\tilde{\boldsymbol{B}}(\boldsymbol{x}^{(i)})^\top\right] \\
&= \tilde{\boldsymbol{B}}(\boldsymbol{x}^{(i)})\boldsymbol{B}(\boldsymbol{x}^{(i)})^\top \text{Cov}\left[\boldsymbol{x}^{(i)}\boldsymbol{x}^{\top(i)}\right]\boldsymbol{B}(\boldsymbol{x}^{(i)})\tilde{\boldsymbol{B}}(\boldsymbol{x}^{(i)})^\top \\
&= \sigma^2 \tilde{\boldsymbol{B}}(\boldsymbol{x}^{(i)})\boldsymbol{B}(\boldsymbol{x}^{(i)})^\top \boldsymbol{B}(\boldsymbol{x}^{(i)})\tilde{\boldsymbol{B}}(\boldsymbol{x}^{(i)})^\top,
\end{aligned}
$$

because the corresponding piecewise linear representation does not change for the small perturbation of the input (Ye & Sung, 2019). Therefore, we have

$$
\tilde{\boldsymbol{B}}(\boldsymbol{x}^{(i)})\boldsymbol{B}(\boldsymbol{x}^{(i)})^\top \simeq \boldsymbol{I},
$$

since $\tilde{\boldsymbol{B}}(\boldsymbol{x}^{(i)})\boldsymbol{B}(\boldsymbol{x}^{(i)})^\top$ is a square matrix. This suggests that

$$
\text{div}_{E_L}\{\boldsymbol{F}(\boldsymbol{x}^{(i)})\} = \sum_i \langle \boldsymbol{b}_i(\boldsymbol{x}^{(i)}), \tilde{\boldsymbol{b}}_i(\boldsymbol{x}^{(i)})\rangle = \text{Tr}\left(\tilde{\boldsymbol{B}}(\boldsymbol{x}^{(i)})\boldsymbol{B}(\boldsymbol{x}^{(i)})^\top\right) \simeq n, \tag{9}
$$

where $n$ is the dimension of the input signal. As the resulting divergence term is just a constant, the contribution of the divergence term is considered trivial and can be neglected.

### 3.2 BAGGING ESTIMATOR

Another important drawback of SURE-based denoising network (4) is that it is difficult to prevent the network from learning a trivial identity mapping. More specifically, if $\boldsymbol{F}_{\boldsymbol{\Theta}}(\boldsymbol{x}) = \boldsymbol{x}$, the cost function in (4) becomes zero. One way to avoid this trivial solution is to guarantee that the divergence term at the optimal network parameter should be negative. However, given that the divergence term comes from the degree of the freedom (Efron, 2004) and the amount of excess optimism in estimating the prediction error (Tibshirani & Rosset, 2018), enforcing negative value may be unnatural. Moreover, with the divergence simplification in (9), the cost function (4) can be simplified as

$$
\ell_{SURE-ED}(\boldsymbol{\Theta}) := \frac{1}{P}\sum_{i=1}^{P}\|\boldsymbol{x}^{(i)} - \boldsymbol{F}_{\boldsymbol{\Theta}}(\boldsymbol{x}^{(i)})\|^2 + 2\sigma^2 n \tag{10}
$$

so that it is much apparent that a trivial solution for (10) is $F_\Theta(x) = x$.

To prevent the neural network from converging to this trivial solution, here we propose a bootstrap aggregation (bagging) scheme (Breiman, 1996). Bagging is a classical machine learning technique which uses bootstrap sampling and aggregation of the results to reduce the variance and to improve the accuracy of the base learner. The rationale for bagging is that it may be easier to train several simple weak learners and combine them into a more complex learner than to learn a single strong learner. More specifically, we use the following bagging estimator:

$$\widetilde{\mu}(x) = \sum_{k=1}^{K} w_k F_\Theta(L_k x), \tag{11}$$

and the corresponding prediction error estimate:

$$\text{Err}(\tilde{\mu}) := E_{x, x^*} \|x^* - \widetilde{\mu}\|^2 = E_{x, x^*} \|x^* - \sum_{k=1}^{K} w_k F_\Theta(L_k x)\|^2,$$

where $\{L_k\}_{k=1}^{K}$ denotes the diagonal matrix whose diagonal elements are either 0 or 1 depending on the bootstrap subsampling patterns, and $\{w_k\}_{k=1}^{K}$ is the corresponding weights. The corresponding empirical prediction error estimate is then given by

$$\ell(\Theta, \{w_k\}) := \frac{1}{P} \sum_{i=1}^{P} \|x^{(i)} - \sum_{k=1}^{K} w_k F_\Theta(L_k x^{(i)})\|^2, \tag{12}$$

Note that $x^{(i)} \neq L_k x^{(i)}$ due to the subsampling patterns so that the optimal neural network $F_\Theta$ that minimizes (12) cannot be the trivial identity mapping. Furthermore, the following proposition shows that the prediction error for the bagging estimator (11) is not bigger than the average of the individual error estimate.

**Proposition 1.** *Let $w_k \geq 0$ and $\sum_k w_k = 1$. Then, we have*

$$\frac{1}{P} \sum_{i=1}^{P} \sum_{k} w_k \|x^{(i)} - F(L_k x^{(i)})\|^2 \geq \frac{1}{P} \sum_{i=1}^{P} \left\| x^{(i)} - \sum w_k F(L_k x^{(i)}) \right\|^2.$$

*Proof.* For any $x \in \{x^{(i)}\}_{i=1}^{P}$, we have

$$\begin{aligned} \sum_{k} w_k \|x - F(L_k x)\|^2 &= x^\top x - 2 x^\top \sum_{k} w_k F(L_k x) + \sum_{k} w_k \|F(L_k x)\|^2 \\ &\geq x^\top x - 2 x^\top \sum_{k} w_k F(L_k x) + \|\sum_{k} w_k F(L_k x)\|^2 \\ &= \left\| x - \sum w_k F(L_k x) \right\|^2, \end{aligned}$$

where we use the Jensen's inequality for the inequality. By summing up for all training data $\{x^{(i)}\}_{i=1}^{P}$, we conclude the proof. ☐

The gap increases more when the $F(L_k x)$ provides diverse output for each realization of the index $L_k$ (Breiman, 1996). In fact, the authors in (Ye & Sung, 2019) showed that the input space results in non-overlapping partitions with different linear representations so that by changing the subsampling pattern $L_k$, the distinct representation may be selected, which can make the corresponding bagging estimate more accurate.

## 4 IMPLEMENTATION DETAILS

The final loss function for our bagging estimator is given by

$$\ell(\Theta, \Xi) := \frac{1}{P} \sum_{i=1}^{P} \|x^{(i)} - \sum_{k=1}^{K} w_k(\Xi) F_\Theta(x, L_k)\|^2, \tag{13}$$

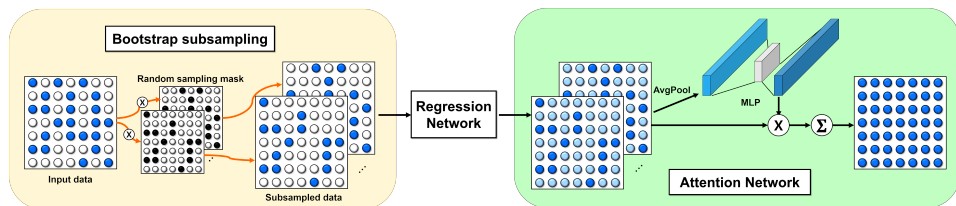

Figure 1: Overall network architecture of the proposed method. The acquired data is split into multiple subset using bootstrap subsampling. The data is then processed using a regression network followed by an aggregation module using an attention network.

where we also parameterize the weighting using a neural network with parameter $\Xi$. Although the standard way of aggregation in bagging estimator is a simple average of the overall results of the regression network, this may not be the best method when the number of bootstrap subsampling is limited. Instead, we propose a weighted average scheme whose weight is calculated by data attention module so that it efficiently combines all data by adaptively incorporating output from various bootstrap sub-sampling patterns. A schematic diagram of the proposed method is illustrated in Fig. 1, which consists of three building blocks: bootstrap subsampling, a regression network, and an attention network to calculate the weight parameters.

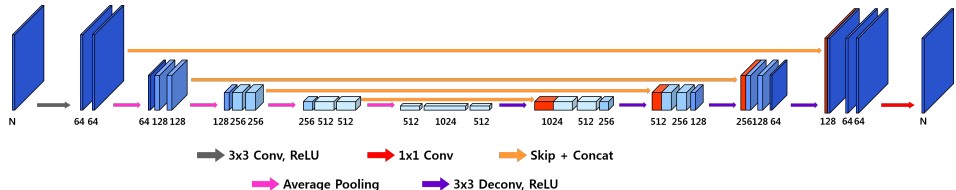

Figure 2: Architecture for the regression network.

In particular, U-net (Ronneberger et al., 2015) in Fig. 2 was used as our regression network. The network is composed of four stages with convolution, batch normalization, ReLU, and skip connection with concatenation. Each stage is composed of three $3 \times 3$ convolution layers followed by batch normalization and ReLU, except for the last layer, which is $1 \times 1$ convolution layer. The number of convolutional filters increases from 64 in the first stage to 1024 in the final stage. To estimate the weight $w_k$, our attention network consists of two fully connected layer. The input dimension of the attention network is $\mathbb{R}^{1 \times 1 \times K}$ followed by the average pooling of the concatenated output of regression network. The number of hidden node is 64, and the final dimension of the output is $\mathbb{R}^{1 \times 1 \times K}$ for aggregation.

The overall network was trained using Adam optimization with the momentum $\beta_1 = 0.9$ and $\beta_2 = 0.999$. The proposed network was implemented in Python using TensorFlow library and trained using an NVidia GeForce GTX 1080-Ti graphics processing unit.

## 5 EXPERIMENTAL RESULTS

Experiments were conducted for various inverse problems such as accelerated magnetic resonance imaging (MRI), energy-dispersive X-ray spectroscopy (EDX) (Solé et al., 2007), and image super-resolution in Appendix.

### 5.1 ACCELERATED MRI

In accelerated MRI, the goal is to recover high quality MR images from sparsely sampled $k$-space data to reduce the acquisition time. This problem has been extensively studied using compressed sensing (Lustig et al., 2007), but recently deep learning approaches have been the main research interest due to its excellent performance at significantly reduced run time complexity (Hammernik et al., 2018; Han & Ye, 2019). A standard method for neural network training for accelerated MRI is based on the supervised learning, where the MR images from fully sampled $k$-space data is used

as reference and subsampled and zero-filled $k$-space data is used as the input for the neural network. Specifically, we use the $R = 13.45$ subsampled k-space data as neural network input and the goal is to obtain high resolution images from the sub-sampled $k$-space data. As a baseline algorithm, the state-of-the-art deep learning algorithm called $k$-space deep learning (Han & Ye, 2019) is employed, which interpolates the missing $k$-data using a CNN.

In the proposed method, the accelerated $k$-space data is further subsampled using bootstrap sampling with the subsampling ratio of 91 %, and the number of bootstrap sample set $K$ was set to 10. The same $k$-space deep learning network (Han & Ye, 2019) was used for our regression network using bootstrap samples. We trained the baseline algorithm and our method under the same conditions, except that the proposed network was trained by minimize the loss (13). The initial learning rate was set to $10^{-2}$, and it was divided by half at every 50 epochs until it reaches to around $10^{-4}$. The training data was generated from Human Connectome Project (HCP) MR dataset (https://db.humanconnectome.org). Among the 34 subject data sets, 28 subject data sets were used for training and validation. The other subject data sets were used for test. We also provide reconstruction results by GRAPPA (Griswold et al., 2002), which is a standard $k$-space interpolation method in MRI.

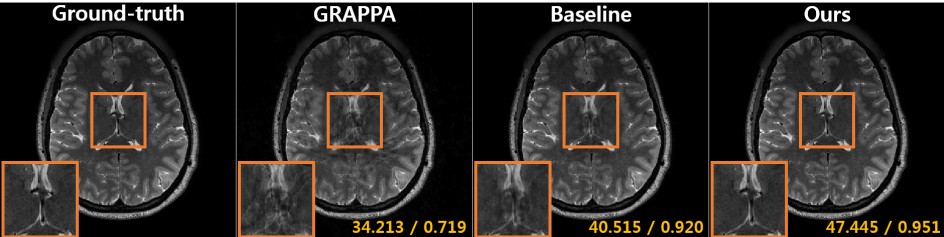

Figure 3: Reconstruction results using accelerated MR data at the acceleration factor of $R = 13.45$. The PSNR and SSIM index values for each images are written at the corner.

Thanks to the bagging, the proposed method provided nearly perfect reconstruction results compared to other algorithms as shown in Fig. 3. Moreover, its average peak signal-to-noise ratio (PSNR) and structural similarity (SSIM) index for entire test dataset significantly outperform existing methods as shown in Table 1.

Table 1: Quantitative comparison using PSNR and SSIM index.

|  | GRAPPA | Baseline (Han & Ye, 2019) | Ours |
|---|---|---|---|
| PSNR (dB) | 34.52 | 38.67 | 42.72 |
| SSIM | 0.72 | 0.89 | 0.91 |

To demonstrate the optimality of the proposed architecture, we performed extensive ablation study. For the ablation study, we used 13 subject data sets from HCP MR dataset, from which seven subject data sets were used for training and validation and the remaining dataset were used for test. The ablation study was performed by excluding some structures from the network and applying the same training processes. First, our theory says that the performance improvement of our method increases with the number of bootstrap subsamples $K$. To verify this, we performed comparative studies with different $K$ values such as 1, 2, 4 and 10. As shown in Table 2, the proposed network provided better reconstruction results as the number of bootstrap sample set $K$ increased. This clearly confirmed the importance of the bootstrap subsampling. Second, our derivation suggests that the batch normalization plays a key role to simplify the divergence term. As shown in Fig. 4 and Table 3, the lack of batch normalization produces a significantly degraded image with an average PSNR drop from 40.047 dB to 25.290 dB. This verified the important role of batch normalization.

Table 2: Quantitative comparison using PSNR and SSIM index with respect to number of bootstrap subsample sets.

| K | 1 | 2 | 4 | 10 |
|---|---|---|---|---|
| PSNR (dB) | 38.68 | 39.81 | 40.05 | 40.27 |
| SSIM | 0.89 | 0.90 | 0.91 | 0.93 |

We also evaluated the importance of the attention network to calculate the weights. With standard bagging, the final output is a simple average of the entire network output for each bootstrap subsamples without the attention network. As shown in Fig. 4 and Table 3, the simple average produced blurry results than by the attention network and the PSNR and SSIM values were worse. This confirmed that the important of attention network for the weight calculation.

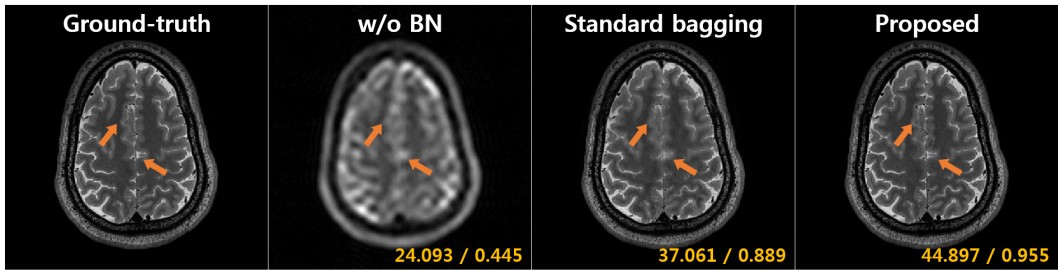

Figure 4: Ablation study for the accelerated MR experiments at the acceleration ratio of $R = 13.45$. The PSNR and SSIM index values for each images are written at the corner. w/o BN: reconstruction results from an ablated network without batch normalization, Standard bagging: reconstruction results from an ablated network without an attention module.

Table 3: Quantitative comparison in terms of PSNR and SSIM index in our ablation study.

|  | without BN | Standard bagging | Ours |
| --- | --- | --- | --- |
| PSNR (dB) | 25.29 | 35.91 | **40.05** |
| SSIM | 0.49 | 0.867 | **0.914** |

## 5.2 ENERGY-DISPERSIVE X-RAY SPECTROSCOPY IMAGING

As for another experiment, we use the energy-dispersive X-ray spectroscopy (EDX) (Solé et al., 2007) data set which were measured by scanning transmission electron microscopy. EDX (Solé et al., 2007) is an analytical technique used for spectroscopic characterization of a sample. It relies on the emission of characteristic X-rays from a specimen that is generated by a high-energy beam of electrons. Specifically, the incident beam may eject an electron from the inner shell of atoms. Then, an electron from an outer shell then fills the hole, and the energy difference may be released in the form of an X-ray. As the energies of the X-rays are characteristic of the energy differences that are determined by the atomic structure of the emitting element, EDX is widely used for nano-scale quantitative and qualitative elemental composition analysis by measuring X-ray radiation from the interaction with high energy electron and the material (McDowell et al., 2012).

However, in EDX, the specimens can be quickly damaged by the high energy electrons, so the acquisition time should be reduced to its minimum. This usually results in very noisy and even incomplete images as shown in Fig. 5(a). Therefore, the goal of this experiment was to remove the noise and reconstruct high resolution EDX images. The main technical difficulty is that there are no noiseless reference data. Due to the lots of missing photons, a widely used approach for EDX analysis is a kernel regression method using an average kernel as shown in Fig. 5(b). Unfortunately, this often results in severe blurring. Existing unsupervised learning methods such as SURE network (Soltanayev & Chun, 2018) and Noise2Noise (Lehtinen et al., 2018) cannot be used for this purpose, since there are no specific noise models for the EDX and the noiseless clean data is not available for Noise2Noise training. In fact, this difficulty of EDX denoising was our original motivation for this work. For our network training and inference, we use bootstrap subsampled images from the measurement image in Fig. 5(a). In addition, we used the measurement image in Fig. 5(a) as the label for training.

We used 28 cases from the EDX dataset. The specimen are composed of quantum-dots, where core and shell consist of Cadmium (Cd), Selenium (Se), Zinc (Zn), and Sulfur (S). The regression network was trained using image patches, whose sizes was $128 \times 128$. For stable training, the intensity of data was normalized between [0, 1]. The initial learning rate was set to $10^{-3}$, and it was divided in

half at every 50 epochs until it reaches around $10^{-5}$. For the bootstrap subsampling, the number of random subsampling mask was $K = 30$. The regression network was optimized to minimize the loss (13) with respect to $\Theta$ first, after which the attention network was trained to properly aggregate the entire interpolated output. As shown in Fig. 5(d), our method successfully produced high resolution images which are significantly better than existing kernel regression methods. Thanks to the bagging procedure, the regression network is believed to learn the measurement statistics so that the network not only removes the noise but also filled in the unmeasured data. We also compared the performance between the standard bagging and the proposed method. As shown in Fig. 5(c) and (d), the weighted averaging from the attention module provides better performance over the the standard bagging that uses the simple average. This confirmed the efficacy of the proposed method.

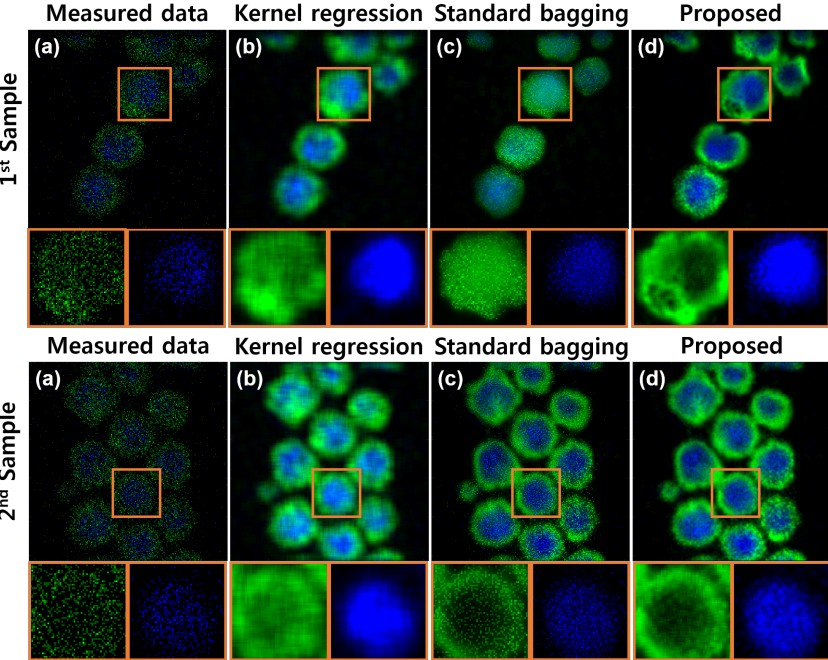

Figure 5: EDX experimental results. Green and blue particles refer to Zn and Cd, respectively. (a) Input data, and the reconstruction results using (b) the existing kernel regression method, and the proposed methods (c) with simple average (standard bagging) and (d) the attention based weighted average.

## 6 CONCLUSION

In this paper, we proposed a novel bagging scheme of encoder-decoder CNNs for various inverse problems. The algorithm was derived by the observation that the existing SURE-based unsupervised denoising networks have several practical limitations due to the divergence term and potential to converge to a trivial solution. Inspired by the recent discovery of basis-like representation of the encoder-decoder CNN, we provide a simple approximation of the divergence term. This also led to a novel bagging scheme to prevent from converging to the trivial identity mapping. In the implementation, multiple input data were generated by bootstrap subsampling, after which final result are obtained by aggregating the entire output of network using an attention network. Experimental results with accelerated MRI and EDX experiments showed that the proposed method provides consistent improvement for various inverse problems.

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

## A   APPENDIX: SUPER-RESOLUTION APPLICATION

Single image super-resolution (SR) is a task to estimate a high-resolution (HR) image from its low-resolution (LR) image. To demonstrate the effectiveness of the proposed method, the proposed bagging scheme was also applied to these problems. Deep Back-Projection Network (DBPN) (Haris et al., 2018) is a recent state-of-the-art method for this applications, so we employed DBPN as our baseline model to improve its performance by the proposed method. The baseline network is trained using DIV2K (Agustsson & Timofte, 2017) with totally 800 training images on the $\times 2$ and $\times 4$ (in both horizontal and vertical directions) super-resolution tasks. The network consists of three modules: initial feature extraction, back-projection stages, and reconstruction. The initial LR feature maps were extracted using the convolution layer of the initial feature extraction module. By applying the sequence of projection units which alters between configuration of LR and HR feature maps, the extensive HR features maps can be obtained. The final HR image was reconstructed using the overall HR feature maps from the projection units.

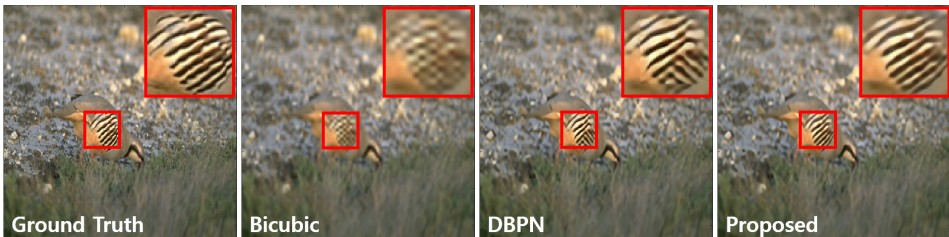

Figure 6: Comparisons between DBPN and the proposed method for $\times 4$ super-resolution task.

Training conditions of the baseline algorithm and our method were followed as described in (Haris et al., 2018). Specifically, we used patch processing whose size is $32 \times 32$ for LR image. The initial learning rate was set to $10^{-4}$, and it was divided by 10 at every $5 \times 10^5$ iterations for total $10^6$ iterations. For the training of our bagging network, the number of random subsampling mask $K$ was set to 32 and 8 for $\times 2$ and $\times 4$ task, respectively. The subsampling ratio for bootstrap sampling was set to $80\%$. In addition, the entire networks were trained simultaneously to minimize the loss (13).

Thanks to the bootstrapping and aggregation using attention network, the data distribution can be fully exploited to restore the high resolution components, which results in the properly reconstructed details of the image as shown in Fig. 6. As described in Table 4, our method can also improve the quantitative performance of the super-resolution task.

Table 4: Comparison of PSNR and SSIM index for super-resolution task

| Algorithm | Scale | set14 | | bsd100 | | Algorithm | Scale | set14 | | bsd100 | |
| | | PSNR | SSIM | PSNR | SSIM | | | PSNR | SSIM | PSNR | SSIM |
|---|---|---|---|---|---|---|---|---|---|---|---|
| DBPN (Haris et al., 2018) | 2 | 30.748 | 0.937 | 31.689 | 0.950 | DBPN (Haris et al., 2018) | 4 | 26.191 | **0.837** | 26.803 | **0.855** |
| Proposed | 2 | **30.802** | **0.939** | **31.720** | **0.951** | Proposed | 4 | **26.222** | 0.837 | **26.855** | 0.855 |

