# OpenReview forum: "BOOSTING ENCODER-DECODER CNN FOR INVERSE PROBLEMS"
_ICLR.cc/2020/Conference — Reject_

### Official Review · AnonReviewer3 · 2019-10-22
**Official Blind Review #3**

**Rating:** 6

**Review:**

1. Summary
The authors address the problem of efficiently employing the SURE estimator as a network training regularizer. They show that for CNN autoencoders this can be efficiently computed. Their other contribution is a bagging/boosting technique which is proved to avoid trivial solutions. The proposed architecture, motivated by the theoretical statements, is shown to outperform classic and 2019 state of the art image reconstruction algorithms in MRI and EDX.
2. Decision and arguments
Unfortunately this paper is outside my expertise so I can’t evaluate the novelty of the theoretical accomplishments. However taking that as a given, they well-motive the proposed architecture and achieve impressive experimental results. The experiments are well described.
3. Questions
a) Why do Table 1 and Figure 3 provide different PSNR and SSIM values?
b) Is there any way to measure accuracy to ground-truth with the EDX data? Or are the results just qualitative?
c) With respect to Figure 2, and in general for autoencoders, the input and output have the same dimension. So how do you reconcile this with undersampled MRI and EDX data? I understand you train on fully sampled data—then how do you input undersampled data? Are the unknown samples set to zero?


**Experience Assessment:**

I do not know much about this area.

**Review Assessment: Checking Correctness Of Derivations And Theory:**

I assessed the sensibility of the derivations and theory.

**Review Assessment: Checking Correctness Of Experiments:**

I assessed the sensibility of the experiments.

**Review Assessment: Thoroughness In Paper Reading:**

I made a quick assessment of this paper.

---

> ### Author Response · Authors · 2019-11-08
> **Reply to Reviewer3: Thanks for the positive comments**
>
> However taking that as a given, they well-motive the proposed architecture and achieve impressive experimental results. The experiments are well described.
>
> ==> Thanks for your understanding. We have also added additional experiments for the super-resolution tasks with natural images in the revised paper. Additionally, the revised paper also provides a comparison with the standard bagging baseline to show that adaptive averaging with the help of the attention module improves the performance. The results consistently show that the proposed method is better than the existing approaches.
>
> 3. Questions
> a) Why do Table 1 and Figure 3 provide different PSNR and SSIM values?
>
> ==> Fig. 3 was the PSNR/SSIM value for the individual figures, whereas Table 1 provides the average values for the *entire* test data set. We have clarified this in the main context in the revised paper.
>
> b) Is there any way to measure accuracy to ground-truth with the EDX data? Or are the results just qualitative?
>
> ==> For the EDX case, no truth-relevant data is available for supervised training. This was the main reason why we developed our method. However, we would like to assure the reviewer that our material scientist has confirmed that the denoised images are clearly consistent with the expected layered structures of the quantum dots that they had expected from their manufacturing protocols.   To provide more experimental results with the groud-truth data, the revised manuscript also provides the additional experimental results for super-resolution tasks. The results clearly show that the proposed method improves the performance.
>
> c) With respect to Figure 2, and in general for autoencoders, the input and output have the same dimension. So how do you reconcile this with undersampled MRI and EDX data? I understand you train on fully sampled data—then how do you input undersampled data? Are the unknown samples set to zero?
>
> ==>  Yes, you are right.  Thanks for your careful observation.  For the MRI case, we use the zero-filled k-space data as our network input, and for the EDX case, we used the noisy image as input since they are the same size as the network output.

---

### Official Review · AnonReviewer1 · 2019-10-23
**Official Blind Review #1**

**Rating:** 1

**Review:**

Summary: The authors consider an encoder decoder setup for linear deblurring problem and propose efficient boosting estimators. Specifically, they use the Stein's unbiased risk estimator for the problem when the noise is gaussian. In the case when the encoder and decoder is represented by a convolutional neural network with RELU activations, they show how they can exploit the recent theoretical results that show the kernel type results to make their procedure efficient. They then propose using a set of models (boosting) and prove that the boosted loss function lower bounds the "nonboosted" loss function.

1. I think Proposition 1 has minor errors, there is no need to apply Jensen's inequality since there's nothing random, but I think the claim is correct -- it is trivial. In experiments, they use attention network which is not a CNN, so I'm not sure how any of the theory applies to this case, can you please clarify?

2. Experimental focus of the paper is to analyze biomedical datasets -- HCP, EDX and the authors compared their method to *only* one baseline. I suggest that they perform some more comparisons on natural images like http://vllab.ucmerced.edu/wlai24/cvpr16_deblur_study/

**Experience Assessment:**

I have published one or two papers in this area.

**Review Assessment: Checking Correctness Of Derivations And Theory:**

I carefully checked the derivations and theory.

**Review Assessment: Checking Correctness Of Experiments:**

I carefully checked the experiments.

**Review Assessment: Thoroughness In Paper Reading:**

I read the paper thoroughly.

---

> ### Author Response · Authors · 2019-11-08
> **Reply to Reviewer1: We have clarified the contents to avoid confusion by readers**
>
> General Comments:
> ==> Thanks for the constructive comments. In the revised article and in this letter we have done our best to clarify the contents of the article and to avoid confusion by the reviewers.
>
> 1.(1) I think Proposition 1 has minor errors, there is no need to apply Jensen's inequality since there's nothing random, but I think the claim is correct -- it is trivial.
>
> ==> We would like to assure the reviewer that Jensen's inequality does NOT necessarily require a random variable since it relates the value of a convex function of an integral (or sum) to the integral (or sum) of the convex function (Please see https://en.wikipedia.org/wiki/Jensen%27s_inequality ).  In fact, Jensen's inequality in the probability theory is a special case of the original Jensen's inequality when a convex function of a random variable is used.
>
> 1.(2)- In experiments, they use attention network which is not a CNN, so I'm not sure how any of the theory applies to this case, can you please clarify?
>
> ==> Thanks for the comment. We would also like to assure the reviewer that the attention module provides only weighting factors $ \ {w_k \} $ for the weighted average calculation in (13), which has nothing to do with the kernel-type results for the encoder-decoder CNN that can simplify the divergence term in the SURE Estimator in (8). Therefore, it does not matter whether it is either in the form of CNN or a fully connected network.  Since we only calculate the K-scalar values in the attention module, the operation is similar to the last layer in a classifier. This led us to use a simple fully connected layer.
>
> 2. Experimental focus of the paper is to analyze biomedical datasets -- HCP, EDX and the authors compared their method to *only* one baseline. I suggest that they perform some more comparisons on natural images like http://vllab.ucmerced.edu/wlai24/cvpr16_deblur_study/
>
> ==> Thanks for the constructive comment.  In our revised manuscript (which has been uploaded),  more comparison results on natural images for the super-resolution task are also provided in the Appendix. Moreover, the comparison results with the standard bagging baseline are also provided. We believe that the additional experimental results clearly showed that the proposed method provides better performance.

---

### Official Review · AnonReviewer2 · 2019-10-24
**Official Blind Review #2**

**Rating:** 3

**Review:**

This paper proposed a piecewise linear close form expression for the Stein’s unbiased risk estimator and use this formulation to construct a new Encoder-decoder convolutional neural network. The author claimed that this closely related to bagging. Improved experimental results on two inverse problems are presented. Overall, the experiment results are encouraging but the paper need clarification on a few points.

1. In the model description part, the intuition behind the attention modules is never mentioned. It will be nice to explain the intuition and possibly attached the derivation of the loss function the attention modules.

2. the author seems misunderstand the difference between boosting and bagging. The way described in the paper is bagging and in order to do boosting, a sequential type of network structure probably need to be proposed.

3. How will be model performance compared with a simple bagging for the baseline compared in the experiment part?


**Experience Assessment:**

I have published one or two papers in this area.

**Review Assessment: Checking Correctness Of Derivations And Theory:**

I assessed the sensibility of the derivations and theory.

**Review Assessment: Checking Correctness Of Experiments:**

I assessed the sensibility of the experiments.

**Review Assessment: Thoroughness In Paper Reading:**

I read the paper at least twice and used my best judgement in assessing the paper.

---

> ### Author Response · Authors · 2019-11-08
> **Reply to Reviewer2: We have addressed your concern**
>
> General Comments:
> ==> We appreciate the reviewer for careful reading and constructive comments.  We have revised the paper accordingly. In particular, we have clearly explained the motivation and provided experimental results that clearly show the advantages of the weighted average from the attention module over the standard bagging baseline.
>
> 1. In the model description part, the intuition behind the attention modules is never mentioned. It will be nice to explain the intuition and possibly attached the derivation of the loss function to the attention modules.
>
> ==>  Thank you for the constructive comments. Although the standard way of aggregation in the bagging estimator is a simple average of the overall results from the regression network, this may not be the best method when the number of bootstrap subsampling is limited. Therefore,   we propose a weighted averaging scheme whose weight is calculated by the data attention module so that it efficiently combines all data by adaptively incorporating output from various bootstrap sub-sampling patterns. As shown in Fig. 4 and Fig. 5, the weighted averaging from the weights calculated by the attention module significantly improved the performance.
>
> 2. the author seems misunderstand the difference between boosting and bagging. The way described in the paper is bagging and in order to do boosting, a sequential type of network structure probably need to be proposed.
>
> ==> Thanks for your careful reading. Although the term "boosting" could be used for general performance improvements, we agree with the reviewer and this revision uses the more accurate term "bootstrap and subsampling (bagging)".
>
> 3. How will be model performance compared with simple bagging for the baseline compared in the experiment part?
>
> ==> Thanks for the suggestion.  Fig. 4 and Fig. 5 now clearly show that the proposed weighted average significantly outperformed the simple bagging baseline.

---

### Author Response · Authors · 2019-11-11
**General Comments to Reviewers**

We thank all reviewers for their careful reviews and constructive comments. The reviewers agreed that the experiment results and the theoretical derivation are encouraging. To address the comment that the paper needs clarification on a few points, we have significantly revised the paper to make the content clears.  The revised paper has been uploaded. The major changes are as follows.

1.  Clarification of attention module

Although the standard way of aggregation in the bagging estimator is a simple average of the overall results from the regression network, this may not be the best method when the number of bootstrap subsampling is limited. Therefore,   we propose a weighted averaging scheme whose weight is calculated by the data attention module so that it efficiently combines all data by adaptively incorporating output from various bootstrap sub-sampling patterns. As shown in Fig. 4 and Fig. 5, the weighted averaging from the weights calculated by the attention module significantly improved the performance.

We also clarified that the attention module is not necessary a CNN to make the theory valid. Note that the attention module provides only weighting factors $\{w_k \} $ for the weighted average calculation in (13), which has nothing to do with the kernel-type results for the encoder-decoder CNN that can simplify the divergence term in the SURE Estimator in (8). Since we only calculate the K-scalar values in the attention module, the operation is similar to the last layer in a classifier. This led us to use a simple fully connected layer.

2. Additional Experiments

In our revised manuscript,  more comparison results on natural images for the super-resolution task are also provided in the Appendix. Moreover, the comparison results with the "standard bagging baseline" are also provided. The additional experimental results clearly showed that the proposed method provides better performance.


3. Other clarification

We have also clarified that PSNR and SSIM values in the table are for the all test data set, whereas those in the figures are for each reconstruction result. In the reply letter, we also pointed out that Jensen's inequality does NOT necessarily require a random variable since it relates the value of a convex function of an integral (or sum) to the integral (or sum) of the convex function.  To avoid potential confusion,  we use the term "bootstrap and subsampling (bagging)" consistently throughout the paper. We have also changed the title accordingly. Other minor typos and grammatical errors have been corrected.

---

### Decision · Program_Chairs · 2019-12-19

**Decision:**

Reject

**Comment:**

This paper introduces a closed-form expression for the Stein’s unbiased estimator for the prediction error, and a boosting approach based on this, with empirical evaluation. While this paper is interesting, all reviewers seem to agree that more work is required before this paper can be published at ICLR.